# Stress Triaxial Constraint and Fracture Toughness Properties of X90 Pipeline Steel

Peng Wang [1,2], Wenqian Hao [3], Jiamiao Xie [3], Fang He [1], Fenghui Wang [1,*] and Chunyong Huo [2,*]

1 Bi-Inspired and Advanced Energy Research Center, Department of Engineering Mechanics, Northwestern Polytechnical University, Xi'an 710129, China; wangpeng013@cnpc.com.cn (P.W.); he_fang@mail.nwpu.edu.cn (F.H.)
2 Tubular Goods Research Institute, China National Petroleum Corporation, Xi'an 710065, China
3 The College of Mechatronic Engineering, North University of China, Taiyuan 030051, China; wqhao@nuc.edu.cn (W.H.); jmxie@nuc.edu.cn (J.X.)
* Correspondence: fhwang@nwpu.edu.cn (F.W.); huochunyong@cnpc.com.cn (C.H.)

**Abstract:** The X90 pipeline steel with high-strength and high-toughness become the most popular pipeline steel. Due to the stress triaxial constraint and fracture toughness properties are the key factors for the stable work of pipeline steel, the research on the fracture toughness of X90 is a great significance to promote the engineering application of high-strength pipeline steel. In order to investigate the stress triaxial constraint and fracture toughness properties of X90 pipeline steel, the experimental rules with different grooves size are proposed using the common toughness experiment and the corresponding numerical models are established in this paper. The resistance curves and fracture toughness of each type of specimens are obtained and compared with that of finite element analysis. Furthermore, the stress distribution, J-integral distribution and stress triaxial constraint of the specimen are analyzed, as well as the influence of side grooves size on the determination of fracture toughness is also discussed. The results obtained from the study will provide a reference to the fracture toughness evaluation research and application of X90 pipeline steel.

**Keywords:** X90 pipeline steel; stress triaxial constraint; fracture toughness; fatigue crack; side grooves; J-integral

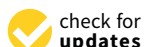



## 1. Introduction

With the advantages of large transportation volume, strong economy and high safety, pipeline transportation has become the main transportation mode of oil and natural gas [1]. Due to the long distance and complex terrain of pipeline transportation, the high toughness, fatigue resistance, corrosion resistance and fracture resistance are the necessary properties of pipeline steel. The research shows that improving the grade of pipeline steel is the most effective way to increase the oil and gas transport capacity and reduce the cost of pipeline construction and transportation [2–4]. The reason for this is that upgrading the grade of pipeline steel, such as X70 to X80, can reduce the amount of pipeline steel by 8% to 12% when the transmission pressure and diameter are determined. At present, X90, X100 and X120 have been successfully trial produced. In fact, X100 and X120 have made very slow progress in industrial application due to their high requirements for strength and toughness. Compared with X80, the strength and toughness of X90 pipeline steel have been improved, and the span is relatively moderate, so it has become an excellent breakthrough point to stride into ultra-high strength steel [5]. Given that the fracture toughness is one of the key factors for the stable work of pipeline steel, the research on the fracture toughness of X90 is of great significance to promote the engineering application of high-strength pipeline steel.

Many investigations have been conducted to study in depth the fracture toughness of various grades of pipeline steels, including the influence of hydrogen, long term ageing, temperature, thickness, pre-strain and so on [6–14]. The static and dynamic fracture

behavior of pipeline steel was analyzed by Luna et al. [15] using Charpy low impact testing and conventional static fracture testing. The relationship between the fracture behavior and strain rate of industrial pipeline steel was also studied. The experimental results showed that the fracture toughness of the material increased slightly with the increase of strain rate. At the different final rolling temperatures, the fracture toughness, microstructure and mechanical properties of two types of API X70 pipeline steel in the transition temperature zone were investigated by Shin et al. [16,17]. In addition, the K-JC curves of pre-split Charpy V-notch specimens were measured using three-point bending test. The influence of pre-strain on the fracture toughness of coiled pipeline steel was discussed by Tkaczyk et al. [18], and the study showed that plastic deformation would reduce the fracture toughness of the material. In addition, the fracture resistance curves of the material in the initial state and the strain state were compared. A new method for determining the fracture toughness of two types of ferritic oil and gas pipeline steels was proposed by Said et al. [19]. The effects of different fatigue crack prefabrication forces on fracture toughness of API X70 pipeline steel were investigated by Nowak-Coventry et al. [20]. The formation mechanism of delamination cracks in API X70 pipeline steel was studied by Park et al. [21] using tensile fracture toughness testing. The delamination cracks of tensile and ductile specimens were shown to be intergranular cracks by observing the microstructure of API X70 pipeline steel. Chanda et al. [22] investigated the influence of temperature on fracture toughness of pipeline steel and proposed a finite element simulation method of a droplet heavy tear test (DWTT) of pipeline steel based on a temperature-dependent cohesive zone model (CZM) to analyze the fracture behavior of pipeline steel at different temperatures. Guo et al. [23,24] investigated the effects of thickness and delamination on the fracture toughness of X60 pipeline steel and found the coupling effect of delamination cracks and out-of-plane stress constraints. With the increase of thickness, the delamination phenomenon is serious, and the three-dimensional stress constraint near the crack tip is almost in the state of low constrained plane stress. Three-point bending specimens of X70 pipeline steel with different thicknesses and initial crack lengths were prepared and tested for fracture toughness at different temperatures by Yang et al. [25] and Dong et al. [26]. The results show that with the decrease of temperature, the specimen changed from ductile failure to brittle failure. In addition, a delamination crack was found on the fracture surface of specimen, and the size and number of cracks were related to the thickness of the specimen, while the location of crack was related to the experiment temperature and the initial crack length. The fracture toughness of X80 pipeline steel at different temperatures through experiment and 3D finite element simulation was investigated by Xu et al. [27]. The results showed that the fracture toughness of X80 pipeline steel decreased obviously with the decrease of temperature, and the fracture type of the specimen gradually tended to brittle fracture. In addition, the real stress-strain curve behavior of the material at different temperatures had good transmissibility from a smooth tensile rod to a fracture mechanics specimen, and temperature had no obvious effect on the hardening behavior of the material. Xiao et al. [28] studied the effect of pre-strain on low temperature fracture toughness of pipeline steel and conducted a tensile experiment and fracture toughness test on X80 pipeline steel raw material and plastic deformation material at different temperatures. The experiment results showed that the yield strength and tensile strength of pipeline steel could be improved by tensile pre-strain, but reduced by compressive pre-strain [10].

Although the side groove size of the fracture toughness test specimen follows a certain standard, the deviations of root radius and side groove depth are generated during the actual testing operation, which affects the determination of fracture toughness. The out-of-plane constraint effect on fracture toughness of the single edge notch tension specimens were experimentally studied by Li et al. [29]. It was found that the critical crack initiation toughness decreases significantly as specimen thickness increases until the thickness-to-width ratio is equal to 4, beyond which thickness, the effect becomes relatively weak. Generally, the experimental determination of material critical crack-tip opening displacement or critical J-integral is based on the standardized procedures recommended by ASTM

E1820 [30] or ISO 12,135 [31], which use the small-scale standardized specimens with deep cracks to guarantee high crack-tip constraint under *J*-dominance scale plasticity, including three-point bending, compact tension and disk-shaped tension specimens. In the process of crack growth in these experiments, the crack usually extends forward in a gradual manner and the trailing edge often fails in shear [32]. Therefore, the crack extension can maintain the straight state of specimen with side grooves because the introduction of side grooves can be regarded as plane strain conditions approximately [32,33]. Some scholars have investigated the effect of side grooves in single-edge bending and compact tension specimens on elastic compliance [32,34], stress intensity factor [32], J-integral and crack-tip constraint by numerical simulation and experimental investigation. The difference between the compliance of a specimen without side grooves and that of a specimen with side grooves (for $B-B_N < 0.25B$) is less than 6% based on the numerical simulation method. The effect of the depth of side grooves on J-integral and constraint parameters for shallow and deep cracks was discussed by Shen et al. [35]. For a theoretical model, a functional dependence of parameter considering side groove, crack size and hardening exponent of CS-19 aluminum alloy was proposed by Sarzosa et al. [36]. However, there are few studies on the influence of side grooves on fracture toughness behavior, stress distribution and J-integral distribution of X90 pipeline steel.

Therefore, in order to investigate the stress triaxial constraint and fracture toughness properties of X90 pipeline steel, the experiment with and without side grooves was taken forward, and for understanding the effect of groove size and stress constraint, the corresponding numerical models are established. The experimental results and simulation results are compared to verify the accuracy of the simulation results. In addition, the stress distribution, J-integral distribution and the stress triaxial constraint distribution of X90 pipeline steel specimen are discussed. The influence of side groove scale on the result of the fracture toughness test is also analyzed. The results obtained from the study will provide a reference to the fracture toughness estimation research and application of X90 pipeline steel.

## 2. Materials and Methods

### 2.1. Experiment Model

The X90 pipeline steel with high-strength and high-toughness has become the most popular pipeline steel. The X90 pipeline steel has been widely used in many fields because of its good weldability, well hydrogen induced cracking resistance and good sulfide stress corrosion cracking resistance. The chemical composition of X90 pipeline steel and the corresponding mass fraction of each element are shown in Table 1. The mechanical properties of X90 pipeline steels are obtained from the uniaxial tensile test of three same X90 bars and their engineering stress vs. engineering strain responses are shown in Figure 1, so the specific values of elastic modulus, Poisson's ratio, yield strength and tensile strength of X90 pipeline steel can be obtained in Table 2.

**Table 1.** Chemical composition of X90 pipeline steel (mass fraction, %).

| Chemical Composition | C | Mn | Si | P | S | Nb | Cu | Ni | Cr | Fe |
|---|---|---|---|---|---|---|---|---|---|---|
| Mass fraction (%) | 0.046 | 1.77 | 0.22 | 0.011 | 0.001 | 0.059 | 0.15 | 0.20 | 0.03 | 97.3 |

**Table 2.** Mechanical properties of X90 pipeline steel.

| Parameters | Density (g/cm³) | Young's Modulus (MPa) | Poisson Ratio | Yield Strength (MPa) | Tensile Strength (MPa) | Total Elongation (%) | $R_{t0.5}/Rm$ [1] |
|---|---|---|---|---|---|---|---|
| Mechanical properties | 7.84 | 206 | 0.3 | 692 | 783 | 13 | 0.88 |

[1] $R_{t0.5}$ is the yield strength (when the total strain is 0.5%) and $Rm$ is tensile strength.

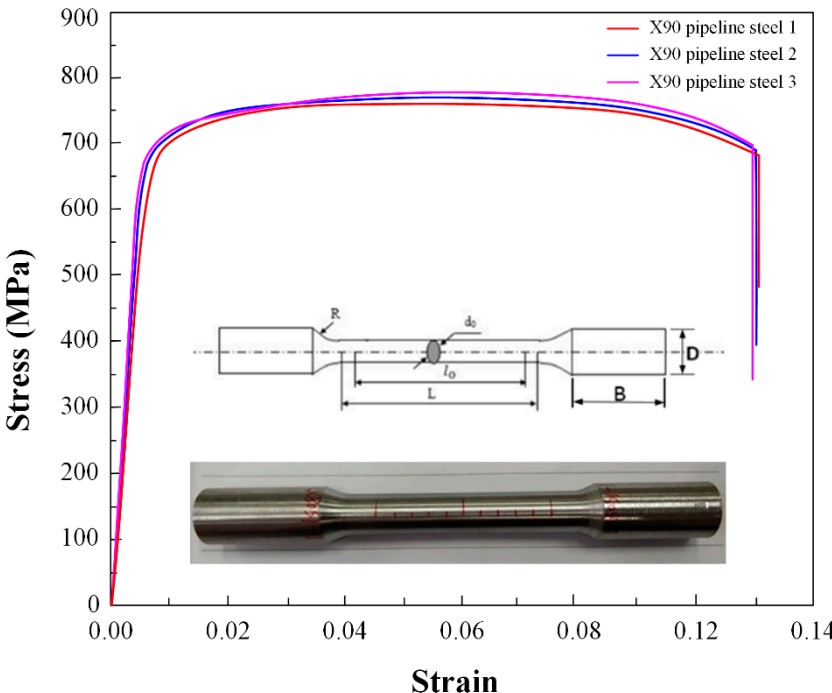

**Figure 1.** Engineering stress vs. engineering strain response obtained by uniaxial tensile test of X90 pipeline steel at room temperature.

The specimens were machined as single edge notched beam (SENB), as shown in Figure 2. The three-point bending experiment was proposed and four different specimen designs were tested in this experiment [37], as shown in Figure 3. The first was a 20 mm thickness ($B$ = 20 mm) specimen without side grooves (SENB 20 (without side grooves)), so the height $W$ and length $S$ of the specimen are $W = 2B = 40$ mm and $S = 4W = 160$ mm respectively (Figure 3a). The second was a 20 mm thickness ($B$ = 20 mm) specimen with side grooves (SENB 20 (with side grooves)), and the depth and angle of side grooves on both sides of specimen was $0.1B$ ($B_N = 0.8B$) and $60°$, respectively (Figure 3b). $B_N$ is the effective thickness of the specimen with side grooves and $B_N = B$ when the specimen without side grooves [31]. The dimensions of the third and fourth specimens are the same as those of the first and second samples respectively, except that the thickness of the third ((SENB 18 (without side grooves))) and fourth (SENB 18 (with side grooves)) specimens is 18 mm ($B$ = 18 mm). The dimensions of four different specimens are shown in Table 3. Six experiments are carried out for each type of specimen. A notch of $0.45W$ mm was initially cut mechanically in the center of the bottom edge of the plate as shown in Figure 2. The micro pre-crack ($0.10W$) was then further prefabricated in the location of notch tip by an Electro-Hydraulic Servo Fatigue Testing Machine (SDS-300, Sinotest Equipment Co., Ltd., Changchun, China) with a 300 KN load cell (accuracy: ±0.5%). The details of fatigue pre-crack is described in Appendix A. Therefore, the length of original fatigue crack is $a_0 = 0.45W + 0.10W$ [31]. The three-point bending experiment was performed using an electronic universal testing machine (AGS-X, Shimadzu Co., Ltd., Kyoto, Japan) with a 100 KN load cell (accuracy: ±0.5%).

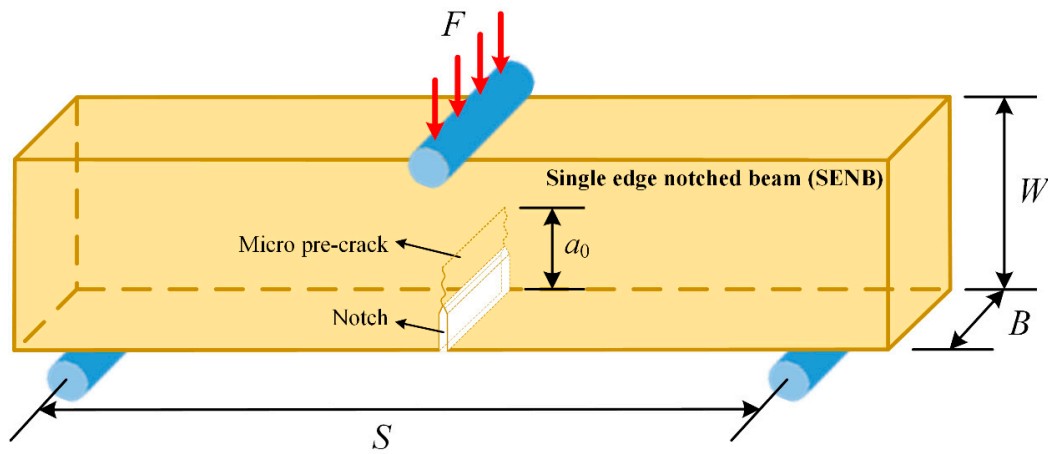

**Figure 2.** Three-point bending experiment of single edge notched beam (SENB).

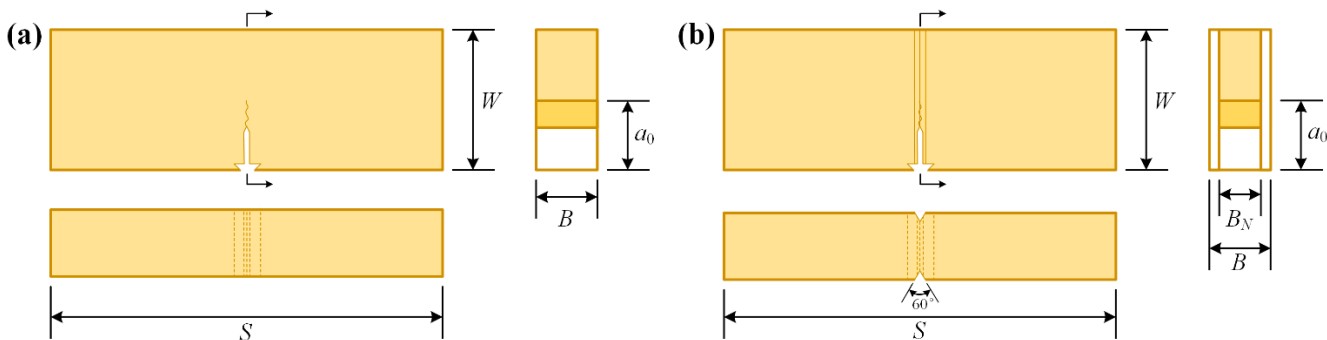

**Figure 3.** Geometric model of two types of specimens: (**a**) specimen without side grooves; (**b**) specimen with side grooves.

**Table 3.** Dimensions of four different specimens.

| Specimen Types | No. | Length $S$ (mm) | Height $W$ (mm) | Thickness $B$ (mm) | Effective Thickness with Side Grooves $B_N$ (mm) [1] |
|---|---|---|---|---|---|
| SENB 20 (without side grooves) | A-1 | 160 | 40.12 | 20.03 | – |
| | A-2 | 160 | 40.14 | 19.94 | – |
| | A-3 | 160 | 40.05 | 19.88 | – |
| | A-4 | 160 | 40.10 | 20.04 | – |
| | A-5 | 160 | 40.12 | 20.11 | – |
| | A-6 | 160 | 40.15 | 20.06 | – |
| SENB 20 (with side grooves) | B-1 | 160 | 40.01 | 20.03 | 16.02 |
| | B-2 | 160 | 40.05 | 20.04 | 16.11 |
| | B-3 | 160 | 40.05 | 19.95 | 16.18 |
| | B-4 | 160 | 40.11 | 20.02 | 16.03 |
| | B-5 | 160 | 40.14 | 20.12 | 16.01 |
| | B-6 | 160 | 40.17 | 20.14 | 15.96 |
| SENB 18 (without side grooves) | C-1 | 144 | 36.01 | 18.04 | – |
| | C-2 | 144 | 36.04 | 18.11 | – |
| | C-3 | 144 | 36.05 | 18.12 | – |
| | C-4 | 144 | 35.98 | 18.14 | – |
| | C-5 | 144 | 36.12 | 18.14 | – |
| | C-6 | 144 | 36.14 | 18.01 | – |

**Table 3.** *Cont.*

| Specimen Types | No. | Length $S$ (mm) | Height $W$ (mm) | Thickness $B$ (mm) | Effective Thickness with Side Grooves $B_N$ (mm) [1] |
|---|---|---|---|---|---|
| SENB 18 (with side grooves) | D-1 | 144 | 36.06 | 17.96 | 14.51 |
| | D-2 | 144 | 36.06 | 17.89 | 14.49 |
| | D-3 | 144 | 36.13 | 18.08 | 14.45 |
| | D-4 | 144 | 36.11 | 18.12 | 14.50 |
| | D-5 | 144 | 36.04 | 18.14 | 14.37 |
| | D-6 | 144 | 36.15 | 18.04 | 14.35 |

[1] $B_N$ is effective thickness of the specimen with side grooves and $B_N = B$ when the specimen without side grooves.

The fatigue crack of the specimen is prefabricated. The minimum prefabricated fatigue crack growth shall be greater than 1.3 mm or 2.5%$W$, whichever is greater. When the crack growth is 1.3 mm, the corresponding maximum fatigue crack prefabricated force is [31]

$$F_f^{1.3} = 0.8 \times \frac{B(W - a_0)^2}{S} \times R_{p0.2} \tag{1}$$

where $B$ is the specimen thickness, $W$ is the specimen width, $S$ is the specimen length, $a_0$ is the initial crack length, $R_{p0.2}$ is the specified non-proportional elongation strength of 0.2% perpendicular to the crack plane at the test temperature.

While when the crack growth is 2.5%$W$, the corresponding maximum fatigue crack prefabricated force is [31,38]

$$F_f^{2.5\%W} = \xi \times E \left[ \frac{(W \times B \times B_N)^{0.5}}{g_1\left(\frac{a_0}{W}\right)} \right] \left( \frac{W}{S} \right) \tag{2}$$

where $\xi = 1.6 \times 10^{-4}$ m$^{1/2}$, $E$ is the Young's modulus of specimen, $B_N$ is effective thickness of the specimen with side grooves and $B_N = B$ when the specimen without side grooves, and $g_1(a_0/W)$ is the stress intensity factor coefficient and its expression is [31,38]

$$g_1\left(\frac{a_0}{W}\right) = \frac{3\left(\frac{a_0}{W}\right)^{0.5}\left[1.99 - \left(\frac{a_0}{W}\right)\left(1 - \frac{a_0}{W}\right)\left(2.15 - \frac{3.93a_0}{W} + \frac{2.7a_0^2}{W^2}\right)\right]}{2\left(1 + \frac{2a_0}{W}\right)\left(1 - \frac{a_0}{W}\right)^{1.5}} \tag{3}$$

The minimum value between Equations (1) and (2) can be taken as the maximum fatigue crack prefabricated force. Therefore, the maximum fatigue crack prefabricated forces of four specimens can be obtained as shown in Table 4. Based on the maximum fatigue crack prefabricated force in Table 4, the loading stress ratio was set to $R = 0.1$, the peak valley value was input and the frequency was set to 8 Hz to realize fatigue crack prefabrication. Then the length of pre-crack was measured to research the preset value (0.10$W$) in real time and the length of original fatigue crack is $a_0 = 0.45W + 0.10W$, as shown in Figure 4a. Three point bending fracture test was carried out after the prefabricated crack was completed, as shown in Figure 4b. The loading rate of the test was controlled to be 0.5 mm/min, the test data were collected every 0.1 s, and the software automatically recorded and saved the applied load $F$ and notch opening displacement $V$. From the experiment datum, the load-displacement ($F$-$V$) curves could be drawn, the notch opening plastic displacement component $V_p$ and area plastic component $A_p$ can be determined [37].

**Table 4.** Maximum fatigue crack prefabricated forces of four specimens.

| Specimen Types | $F_f^{1.3}$ (kN) (Equation (1)) | $F_f^{2.5\%W}$ (kN) (Equations (2) and (3)) | Minimum of Maximum Fatigue Crack Prefabricated Force $F_f$ (kN) |
|---|---|---|---|
| SENB20 (without side grooves) | 22.4 | 10.5 | 10.5 |
| SENB20 (with side grooves) | 22.4 | 9.4 | 9.4 |
| SENB18 (without side grooves) | 20.2 | 9.0 | 9.0 |
| SENB18 (with side grooves) | 20.2 | 8.0 | 8.0 |

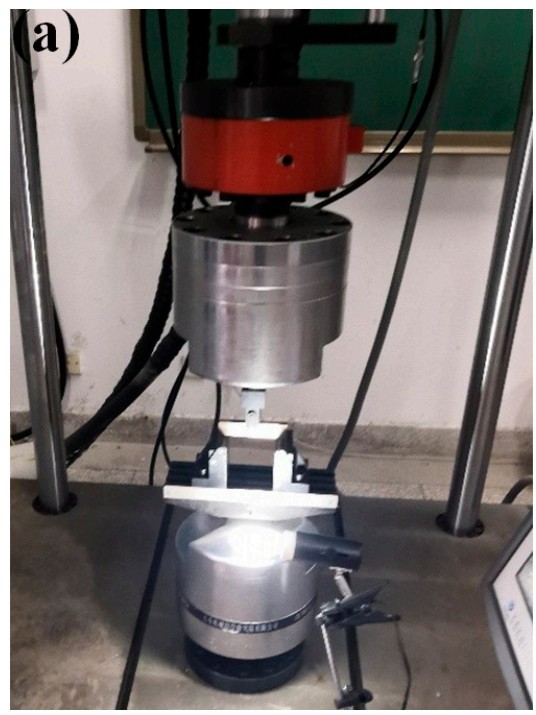 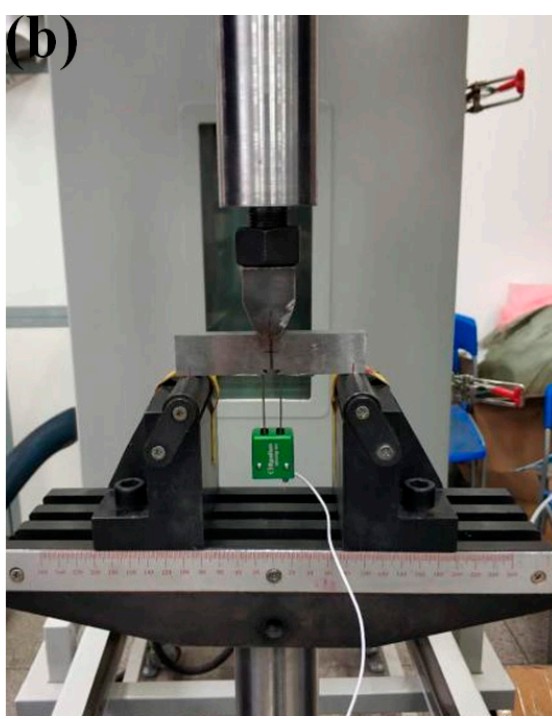

**Figure 4.** (**a**) Micro pre-crack (0.10 W) was prefabricated in the location of notch tip by an Electro-Hydraulic Servo Fatigue Testing Machine (SDS-300) with a 300 KN load cell; (**b**) three-point bending experiment was performed using an electronic universal testing machine (AGS-X) with a 100 KN load cell.

In addition, the fracture toughness can be characterized by J-integral theory. The J-integral values can be calculated by [31,38,39]

$$J = \left[ \frac{FS}{(BB_N)^{0.5}W^{1.5}} \times g_1\left(\frac{a}{W}\right) \right]^2 \left[ \frac{(1-v^2)}{E} \right] + \left[ \frac{\eta_P A_p}{B_N(W-a)} \right] \left\{ 1 - \left[ \frac{\Delta a}{2(W-a)} \right] \right\} \quad (4)$$

where $a = a_0 + \Delta a$, $\Delta a$ is the stable crack growth including blunting, $g_1(a/W)$ could be expressed by Equation (3) except for replacing $a_0$ with $a$. $v$ and $E$ are the Poisson's ratio and the Young's modulus, respectively. $F$ is the applied load. $A_p$ is the area plastic component. The coefficient $\eta_P$ can be determined by

$$\eta_P = 3.667 - 2.119(a_0/W) + 0.437(a_0/W)^2 \quad (5)$$

The curve fitting was carried out for J-integral values of each specimen, and the intersection of fitting curve and passivation line was taken as the fracture toughness value $J_{Q0.2BL}$ of the material. The expression of the fitting curve is [31,38–40]

$$J = \alpha(\Delta a)^\beta \tag{6}$$

where $\alpha$ and $\beta$ are the fitting constants.

For the selection of passivation line form, considering that there is no obvious yield platform in the stress-strain curve of X90 pipeline steel, and in order to obtain relatively conservative results, the expression of passivation line is [32,33]

$$J = 3.75 R_m \Delta a \tag{7}$$

where $R_m$ is the tensile strength of material perpendicular to crack plane at test temperature.

### 2.2. Numerical Simulation Model

The finite element models consistent with the size of experimental specimens are established by ABAQUS software, i.e., 20 mm or 18 mm thick specimens with or without side grooves, which is shown in Figure 5a–d. The isotropic elastic-plastic and homogenized model is employed to describe the specimen material. The nominal stress-strain curve of material is obtained by the tensile test of X90 bar as shown in Figure 1. However, for an accurate and comprehensive description of the strain-hardening behavior of the material, a true stress-strain curve is required. Therefore, according to the relationship between nominal stress $\sigma_{nom}$ (nominal strain $\varepsilon_{nom}$) and real stress $\sigma$ (strain $\varepsilon$), i.e.,

$$\sigma = \sigma_{nom}(1 + \varepsilon_{nom}) \tag{8}$$

$$\varepsilon = \ln(1 + \varepsilon_{nom}) \tag{9}$$

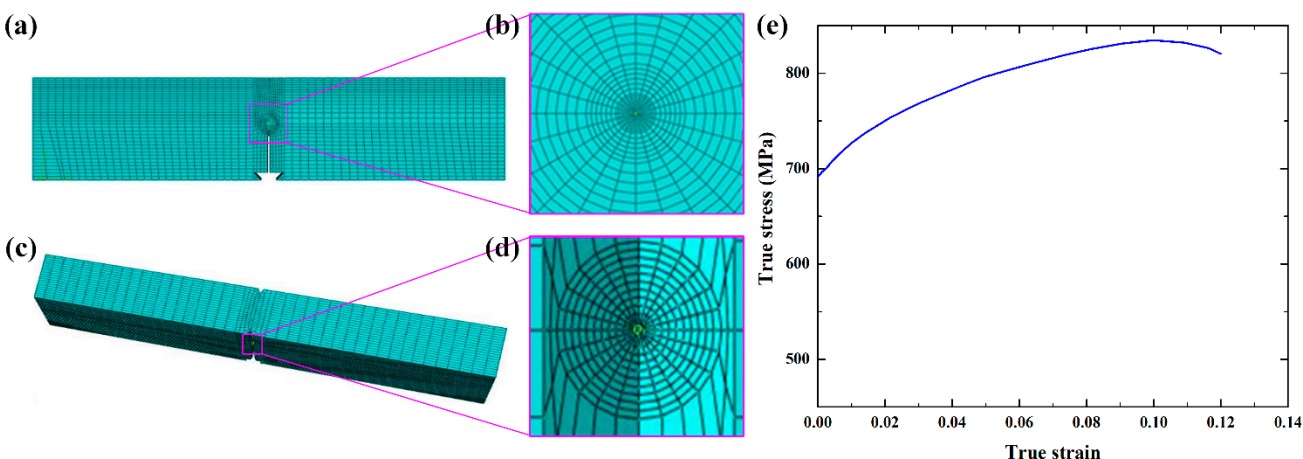

**Figure 5.** Finite element models of specimens: (**a**) overall model without side grooves; (**b**) crack tip of specimen without side grooves; (**c**) overall model with side grooves; (**d**) crack tip of specimen with side grooves; (**e**) relationship between true stress $\sigma$ and true strain $\varepsilon$ calculated by Equations (8) and (9).

The relationship between true stress $\sigma$ and true strain $\varepsilon$ of X90 pipeline steel shown in Figure 5e can be obtained, which can be input into the finite element model as the constitutive relationship, and then the fracture behavior of the material can be simulated. In the finite element model, the elastic-plastic material parameters are shown in Table 5.

The crack is simulated by assigning joints. Taking the crack tip as the center and 0.1 mm, 0.4 mm, 1 mm and 4 mm as the radius of the circle to segment the model. Assign the joint on the crack surface, select the crack front to define the crack propagation direction, set the singular parameter of the crack tip element to 0.25. In order to make the calculation

result as accurate as possible and reduce unnecessary calculation, the grid size should be relatively large at the far end of the crack, while compact at the key part of the crack with the minimum unit located at the crack tip and the size of 0.025 mm. The crack tip meshes of two types of finite element models are shown in Figure 5b,d.

**Table 5.** Elastoplastic material parameters in the finite element model.

| Plastic Strain | 0 | 0.002 | 0.004 | 0.008 | 0.010 | 0.015 | 0.020 | 0.030 | 0.040 | 0.070 | 0.080 | 0.090 | 0.100 | 0.120 |
|---|---|---|---|---|---|---|---|---|---|---|---|---|---|---|
| Yield stress (MPa) | 692 | 701 | 707 | 719 | 728 | 737 | 754 | 769 | 789 | 709 | 827 | 834 | 831 | 828 |

## 3. Results and Discussion

### 3.1. Comparison between Experimental and Simulation Results

Six experiments are carried out for each type of specimens, a series of load-displacement (*F-V*) curves in notch are obtained for each group of specimens, as shown in Figure 6. It can be seen that the coincidence degree of curve at the same specimen group is high, indicating that there is no abnormal specimen in each specimen group. In addition, the uniform reduction of the maximum opening displacement of the notch can also make the distribution of crack propagation $\Delta a$ more uniform, thus facilitating the drawing of J-integral resistance curve.

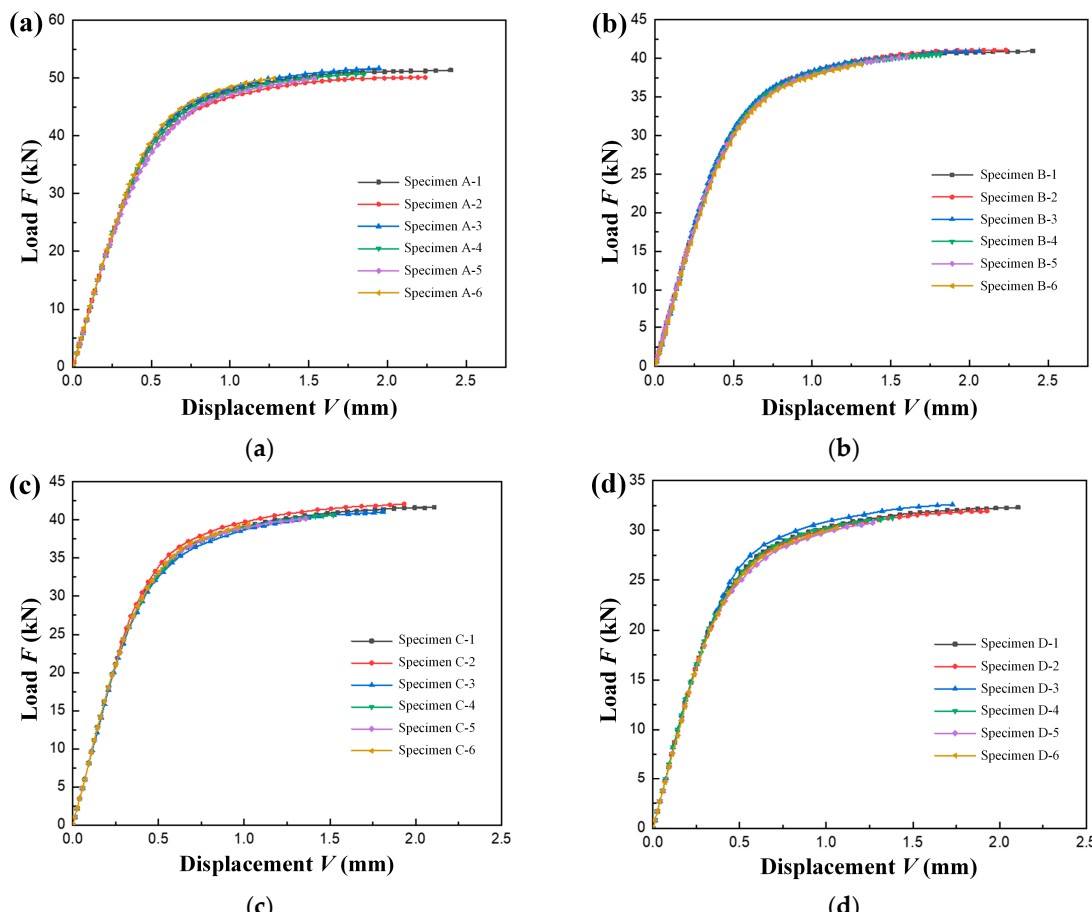

**Figure 6.** Load-displacement (*F-V*) curves of notch for each group of specimens: (**a**) SENB20 without side grooves; (**b**) SENB20 with side grooves; (**c**) SENB18 without side grooves; (**d**) SENB18 with side grooves.

The crack growth $\Delta a$ of each specimen is measured after experiment. Combined with the initial crack length $a_0$ and load-displacement *F-V* curve of the specimen, the J-

integral value corresponding to each specimen is calculated at room temperature according to Equation (4), as shown in Table 6. The fracture toughness can be characterized by J-integral value.

**Table 6.** Fracture toughness characterized by J-integral value of each specimen at room temperature.

| Specimen Types | No. | $a_0$ (mm) | $\Delta a$ (mm) | $g_1$ (a/W) | F (kN) | $A_p$ (J) | J (kJ·m$^{-2}$) |
|---|---|---|---|---|---|---|---|
| SENB20 (without side grooves) | A-1 | 21.75 | 1.16 | 3.40 | 51.26 | 48.11 | 419 |
| | A-2 | 21.66 | 1.04 | 3.34 | 51.28 | 44.67 | 392 |
| | A-3 | 21.54 | 0.82 | 3.24 | 51.04 | 37.80 | 330 |
| | A-4 | 21.63 | 0.70 | 3.22 | 51.12 | 33.21 | 288 |
| | A-5 | 21.72 | 0.47 | 3.19 | 50.85 | 22.91 | 197 |
| | A-6 | 21.71 | 0.18 | 3.12 | 50.42 | 8.59 | 75 |
| SENB20 (with side grooves) | B-1 | 21.84 | 1.25 | 3.47 | 40.86 | 46.96 | 418 |
| | B-2 | 21.93 | 1.08 | 3.43 | 41.03 | 43.52 | 379 |
| | B-3 | 21.95 | 0.84 | 3.36 | 40.82 | 33.21 | 289 |
| | B-4 | 21.76 | 0.66 | 3.25 | 40.65 | 29.78 | 258 |
| | B-5 | 21.81 | 0.43 | 3.21 | 40.55 | 19.47 | 171 |
| | B-6 | 21.86 | 0.21 | 3.15 | 40.34 | 9.28 | 81 |
| SENB18 (without side grooves) | C-1 | 19.68 | 0.98 | 3.41 | 41.62 | 46.96 | 408 |
| | C-2 | 19.62 | 0.84 | 3.35 | 41.63 | 41.24 | 360 |
| | C-3 | 19.71 | 0.63 | 3.31 | 41.98 | 34.36 | 298 |
| | C-4 | 19.73 | 0.57 | 3.29 | 41.71 | 32.07 | 281 |
| | C-5 | 19.59 | 0.34 | 3.18 | 41.47 | 20.62 | 182 |
| | C-6 | 19.60 | 0.17 | 3.13 | 41.12 | 9.16 | 80 |
| SENB18 (with side grooves) | D-1 | 19.81 | 0.95 | 3.43 | 32.25 | 38.94 | 340 |
| | D-2 | 19.77 | 0.75 | 3.37 | 32.01 | 29.78 | 258 |
| | D-3 | 19.74 | 0.64 | 3.32 | 32.32 | 28.63 | 247 |
| | D-4 | 19.68 | 0.51 | 3.25 | 32.04 | 24.05 | 209 |
| | D-5 | 19.72 | 0.36 | 3.22 | 31.87 | 18.32 | 160 |
| | D-6 | 19.83 | 0.19 | 3.21 | 31.64 | 8.93 | 78 |

The J-integral fitting resistance curves and passivation lines at 0.2 mm of SENB20 and SENB18 specimens are shown in Figure 7 based on Equations (6) and (7). It can be seen that there is a certain difference between the J-integral fitting resistance curves measured by the specimens with and without side grooves of the same size, and the difference increases with the increase of crack growth $\Delta a$. The value of intersection point between the J-integral fitting resistance curves and the passivation lines of specimens is the values of $J_{0.2BL}$. The J-integral fitting equations of SENB20 and SENB18 specimens with side grooves are $J = 346.79(\Delta a)^{0.84}$ (red line in Figure 7a) and $J = 348.52(\Delta a)^{0.82}$ (red line in Figure 7b), respectively. The fracture toughness $J_{Q0.2BL}$ of SENB20 and SENB18 specimens with side grooves, which are the intersection points between the J-integral fitting resistance curves (red line in Figure 7) and the passivation lines (green line in Figure 7), are $J_{Q0.2BL} = 102$ kJ·m$^{-2}$ and $J_{Q0.2BL} = 108$ kJ·m$^{-2}$, respectively (Table 7). It can conclude that the J-integral fitting resistance equation of SENB20 specimen with side grooves is relatively close to that of SENB18 specimen, and the fracture toughness value is also relatively close. By contrast, The J-integral fitting equations of SENB20 and SENB18 specimens without side grooves are $J = 377.27(\Delta a)^{0.85}$ (black line in Figure 7a) and $J = 426.35(\Delta a)^{0.79}$ (black line in Figure 7b), respectively. The fracture toughness $J_{Q0.2BL}$ of SENB20 and SENB18 specimens without side grooves, which are the intersection points between the J-integral fitting resistance curves (black line in Figure 7) and the passivation lines (green line in Figure 7), are $J_{Q0.2BL} = 107$ kJ·m$^{-2}$ and $J_{Q0.2BL} = 126$ kJ·m$^{-2}$, respectively (Table 7). The resistance curve fitting equation of SENB20 specimen without side grooves is relatively different from that of SENB18 sample, and the fracture toughness value is also relatively different.

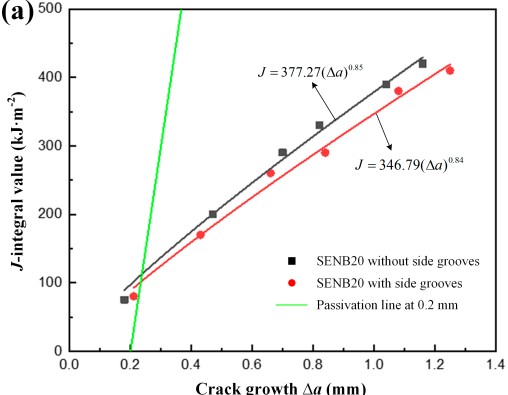
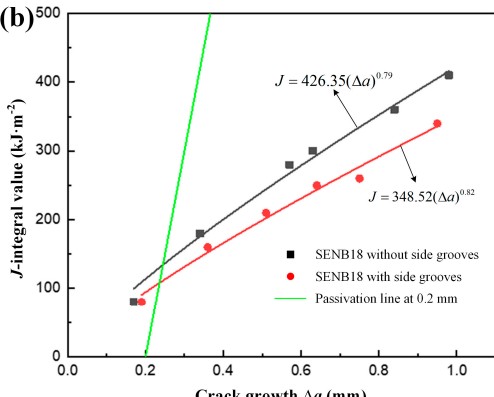

**Figure 7.** J-integral fitting resistance curves and passivation lines at 0.2 mm of SENB20 and SENB18 specimens: (**a**) SENB20; (**b**) SENB18.

**Table 7.** Comparison of fracture toughness values obtained from experiment ($J_{Q0.2BL}$) and simulation ($J$) results.

| Specimen Types | Experiment Result $J_{Q0.2BL}$ (kJ·m$^{-2}$) | Simulation Result $J$ (kJ·m$^{-2}$) | Error (%) |
|---|---|---|---|
| SENB20 (without side grooves) | 107 | 112.5 | 4.21 |
| SENB20 (with side grooves) | 102 | 96.2 | 5.69 |
| SENB18 (without side grooves) | 126 | 120.8 | 4.13 |
| SENB18 (with side grooves) | 108 | 104.1 | 3.61 |

The fracture toughness values of each specimen obtained from simulation ($J$) and experiment ($J_{Q0.2BL}$) results are compared in Table 7. It can be seen that there is little difference between the simulation and experimental results. The errors of the four samples are 4.21%, 5.69%, 4.13% and 3.61%, respectively, which are all within the acceptable range. Therefore, the finite element model employed in this paper can accurately simulate the fracture toughness experiment.

*3.2. Overall Stress Distribution of Specimens*

Take the SENB20 as the example, the overall stress distributions of the specimens SENB20 without side grooves and SENB20 with side grooves are shown in Figure 8. It can be seen that the side grooves have little influence on the overall stress distribution of the specimen. The stress distribution is similar in the area far from the side grooves and slightly different near the side grooves, that is, the maximum stress is concentrated near the crack surface. Therefore, it is necessary to further analyze the stress distribution on the crack surface of two different specimens.

The stress distribution on the crack surface of the specimens SENB20 without side grooves and SENB20 with side grooves at different times are shown in Figure 9. For the specimen SENB20 without side grooves, as shown in Figure 9a, the maximum principal stress at the center of the thickness firstly reaches the maximum at $t_1$ time, and it can be approximated that the specimen starts to crack at this time, that is, the specimen SENB20 without side grooves starts to crack at the center. When the time comes to $t_2$, the crack initiation point gradually propagates from the center to both sides, and the crack growth at the center also increases gradually, as shown in Figure 9c. When the time comes to $t_3$, the cracks along the thickness direction are almost completely generated, but the crack growth varies greatly, i.e., the crack growth at the center of the thickness is much larger than that at the edge. In addition, the crack propagation is like an arch bridge, as shown in Figure 9e. For the specimen SENB20 with side grooves, as shown in Figure 9b, the maximum principal stress value at the edge of the thickness firstly reaches the maximum at $t_1$ time, and it can also be approximated that the specimen starts to crack at this time. It indicated that the

specimen SENB20 with side grooves starts to crack at the edge and slowly propagates to the center. At the $t_2$ time, it can be seen that the center of specimen also cracks rapidly and propagates from the center to both sides, as shown in Figure 9d. At the $t_3$ time, the crack is completely opened along the thickness direction of the whole specimen, and the crack propagates steadily forward. There is little difference in the crack growth along the thickness direction, and the crack front is relatively flat, as shown in Figure 9f. The experiment results of SENB20 without and with side grooves are shown in Figure 9g,h, respectively. The maximum displacement in the thickness center of SENB20 without side grooves are greater than that of SENB20 with side grooves obviously. Therefore, the fracture behaviour of specimens in numerical simulation (Figure 9e,f) agree well with that in the experiment (Figure 9g,h).

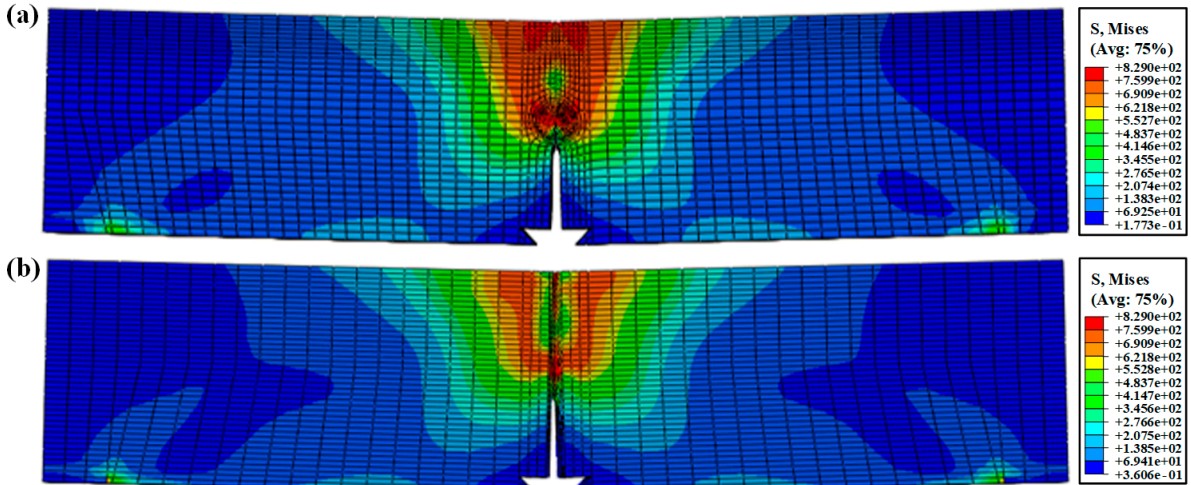

**Figure 8.** Overall stress distributions of the specimens: (**a**) SENB20 without side grooves; (**b**) SENB20 with side grooves.

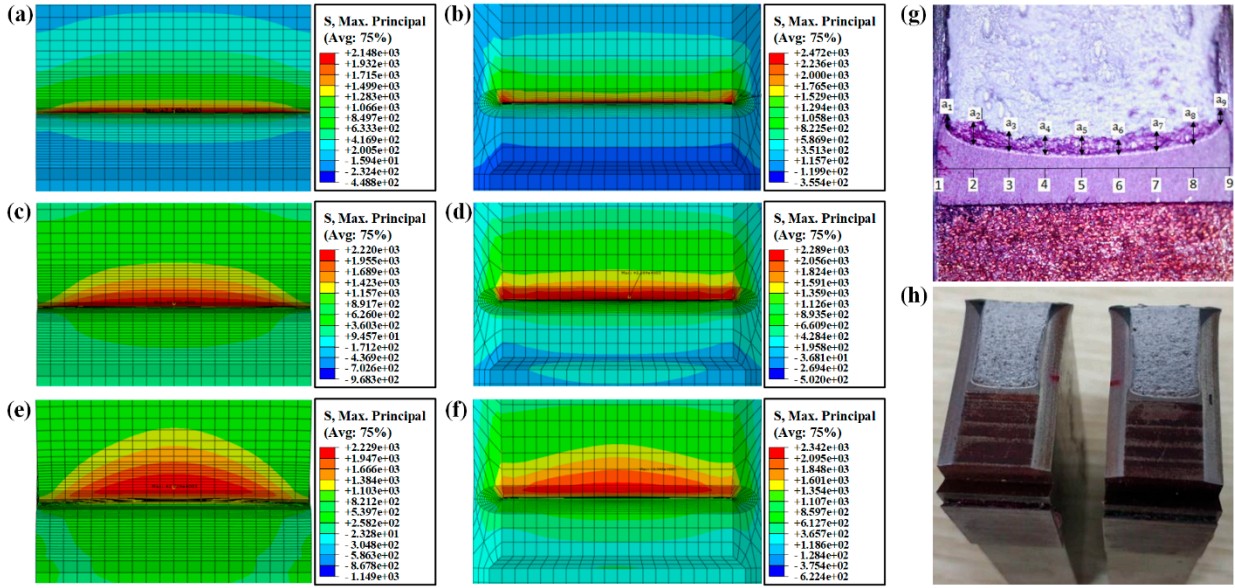

**Figure 9.** Stress distribution on the crack surface of the specimens SENB20 without side grooves and SENB20 with side grooves at different times: (**a**) SENB20 without side grooves at $t_1$ time; (**b**) SENB20 with side grooves at $t_1$ time; (**c**) SENB20 without side grooves at $t_2$ time; (**d**) SENB20 with side grooves at $t_2$ time; (**e**) SENB20 without side grooves at $t_3$ time; (**f**) SENB20 with side grooves at $t_3$ time; (**g**) experiment result of SENB20 without side grooves; (**h**) experiment result of SENB20 with side grooves.

It can be seen from Figure 9a–f that the side grooves have the effect on the stress distribution and crack propagation of specimens. For the specimen without side grooves, the stress of the specimen reaches the maximum firstly in the thickness center and then begins to crack. However, for the specimen with side grooves, the stress of the specimen reaches the maximum firstly in the thickness edge and begins to crack, and the crack at the thickness center starts to crack later and propagates to both sides. In addition, for the specimen without side grooves, the crack growth in the thickness center is large while at the edges on both sides is small. However, for the specimen with side grooves, the crack propagation along the thickness direction is almost the same.

### 3.3. J-Integral Distribution of Specimens

The contour integral method is used to calculate the J-integral distribution of specimens without and with side grooves. According to different integration paths, the integration curve of the specimen in the whole loading process is obtained by creating history output. The fifteen integration paths are compared to selected the most suitable path and the J-integral value of the specimen SENB20 without side grooves for these paths are shown in Figure 10a. Based on the J-integral definition, the path around and closely containing the plastic region of crack propagation is selected to characterize the J-integral value. The integration path $O$ are selected to calculate the J-integral value by observing each stress distribution path of the specimen SENB20, as shown in Figure 10b.

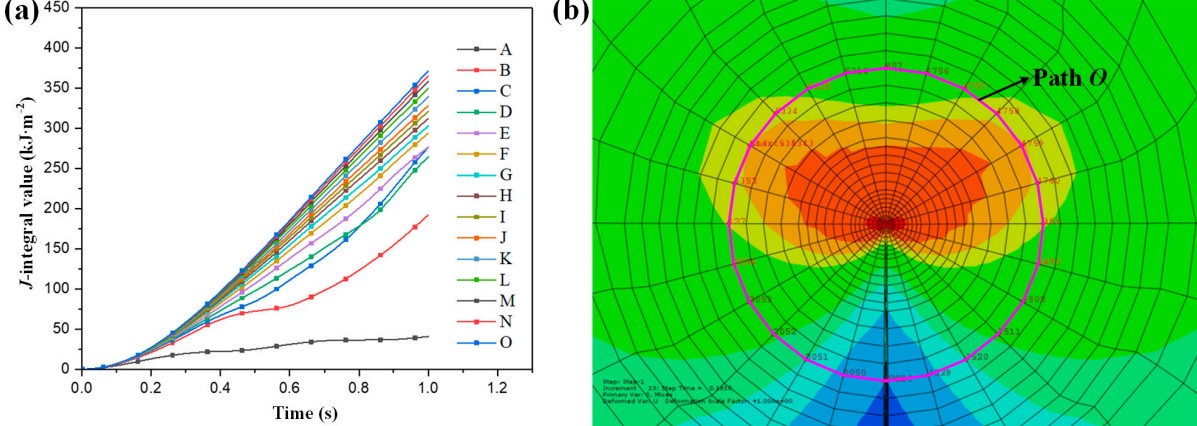

**Figure 10.** J-integral value of the specimens SENB20 without side grooves: (**a**) J-integral value of fifteen integration paths; (**b**) integration path $O$.

As can be seen from Figure 10b, the J-integral distribution along the thickness direction of the specimen is not consistent. The J-integral curves from the center to the edge along the thickness direction of specimens SENB20 with and without side grooves are shown in Figure 11. Where $z/B_N$ represents the position of the specimen along the thickness direction, $z/B_N = 0$ represents the center of the specimen and $z/B_N = 0.5$ represents the thickness edge, i.e., the outer surface of the specimen. It can be seen from Figure 11a that there is a large difference for J-integral value along the thickness direction of the specimen SENB20 without side grooves. The maximum value is located in the thickness center of the specimen ($z/B_N = 0$) and the minimum value is located at the edge of the specimen ($z/B_N = 0.5$). However, there is a small difference for J-integral value along the thickness direction of the specimen SENB20 with side grooves, as shown in Figure 11b. The largest difference is at the edge of the specimen ($z/B_N = 0.5$) and the little difference at the middle of the specimen. Therefore, the J-integral value of specimen without side grooves more sensitive to path selection than those with side grooves. That is, the specimen without side grooves has path-dependence. For the specimen without side grooves, the integration path of the J-integral must be selected; while for the specimen with side grooves, the integration path of the J-integral does not need to be selected during numerical simulation.

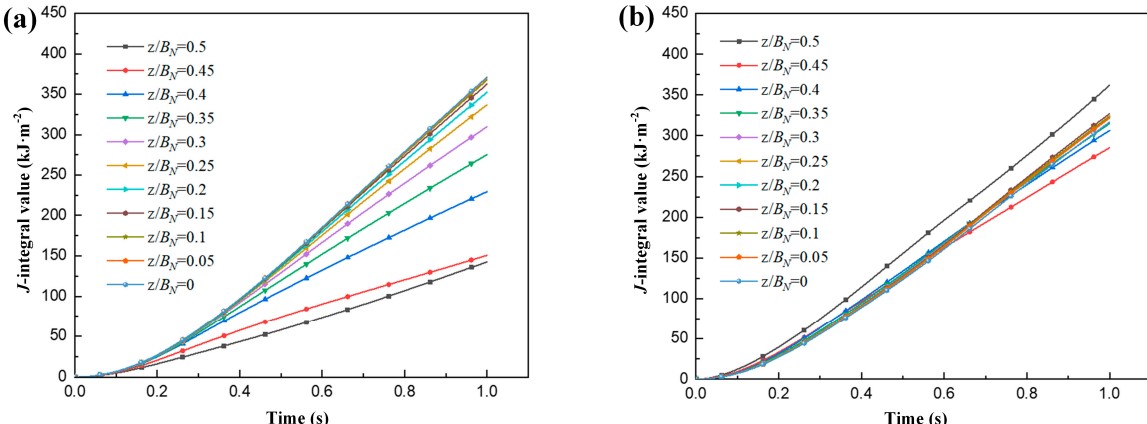

**Figure 11.** J-integral values from the center to the edge along the thickness direction: (**a**) SENB20 without side grooves; (**b**) SENB20 with side grooves.

Due to the J-integral measured in the experiment is the average value, the two specimens can be compared when the J-integral average value $J_{ave}$ is equal. So the J-integral value $J_{ave} = 99$ kJ/m$^2$ of the specimens SENB20 without and with side grooves along the thickness direction is selected to compare (as shown in Figure 12). The comparison in Figure 12 shows the influence of side grooves on the J-integral distribution along the thickness direction of specimen. For the specimen SENB20 without side grooves, the J-integral value along the thickness direction decreases from the center to the edge and the difference reaches 65 kJ/m$^2$. For the specimen SENB20 with side grooves, the J-integral along the thickness direction is basically unchanged and only increases at the edge where the difference is only 26 kJ/m$^2$. Therefore, the J-integral distribution of the specimen along the thickness direction will be more uniform when the side grooves are introduced in the specimen.

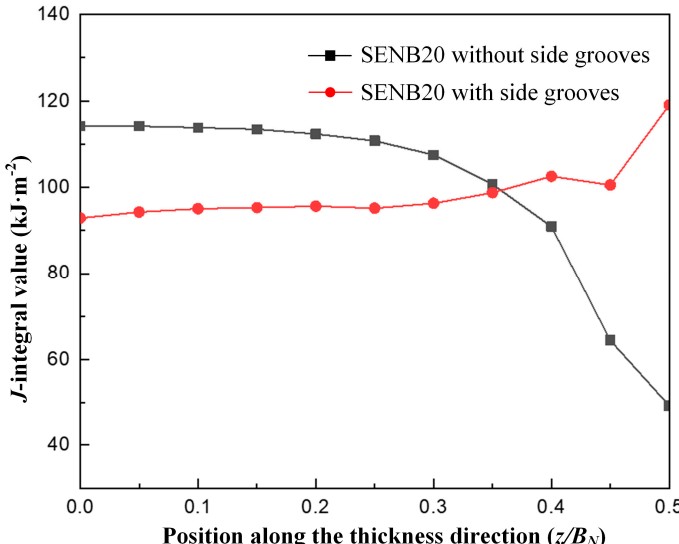

**Figure 12.** J-integral distribution of the specimens SENB20 without and with side grooves along the thickness direction.

### 3.4. Stress Triaxial Constraint Distribution of Specimens

The side grooves change the net thickness of the specimen, which will affect the out-plane constraint of the specimens, so a constraint parameter is introduced to investigate this effect. The out-plane constraint parameter $T_z$, also known as stress triaxial constraint,

is introduced by Guo [41] to analyze the three dimensional out-plane constraint effect. The stress triaxial constraint $T_z$ is a dimensionless parameter and its expression is [41]

$$T_z = \frac{\sigma_{33}}{\sigma_{11} + \sigma_{22}} \tag{10}$$

where $\sigma_{11}$, $\sigma_{22}$ and $\sigma_{33}$ are the normal stresses along $x$, $y$ and $z$ directions, respectively.

When comparing the stress triaxial constraints $T_z$ of different specimens, it should also be carried out when the J-integral average value $J_{ave}$ is equal, where $J_{ave}$ = 99 kJ/m². Through the simulation results, the values of $\sigma_{11}$, $\sigma_{22}$ and $\sigma_{33}$ in the thickness direction of the specimens with and without side grooves can be found out when $J_{ave}$ = 99 kJ/m². Then the stress triaxial constraint $T_z$ in the thickness direction of the specimen can be obtained by substituting the stresses $\sigma_{11}$, $\sigma_{22}$ and $\sigma_{33}$ into Equation (10), as shown in Figure 13.

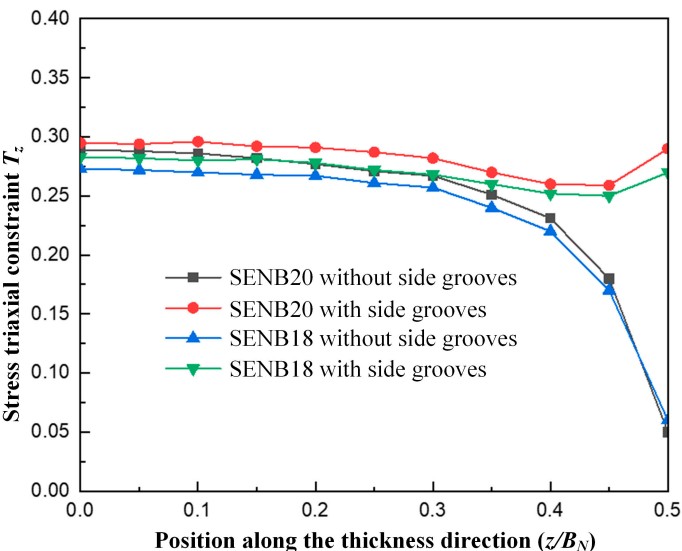

**Figure 13.** Stress triaxial constraint $T_z$ along the thickness direction of the specimen.

The influence of the side grooves on the triaxial stress constraint $T_z$ along the thickness direction of the specimen are shown in Figure 13. For the specimen without side grooves, stress triaxial constraint gradually decreases from the center to the edge, and the overall constraint distribution gradient is large. The variation range of stress triaxial constraint is very small near the center of specimen, while it decreases sharply and tends to 0 near the edge. For the specimen with side grooves, stress triaxial constraint change little along the whole thickness direction, and the overall constraint distribution is relatively gentle. The higher the constraints, the more difficult to produce plastic deformation and the easier it is to crack propagation, thus the difficulty of crack propagation along the thickness direction is different between the two specimens with and without side grooves. Therefore, the crack propagation of the specimen with side grooves is relatively flat as a whole (as shown in Figure 9f), while the crack propagation of the specimen without side grooves is arch bridge type (as shown in Figure 9e). This conclusion is consistent with that of the Section 3.2.

For the specimens with different thicknesses ($B$ = 20 mm and $B$ = 18 mm), the variation trends of stress triaxial constraint along the thickness direction are same. Compared with the specimens without side grooves, the stress triaxial constraint at the center of thickness of SENB20 and SENB18 specimens with side grooves increases by 2.1% and 3.6%, respectively. Therefore, the side grooves can improve the stress triaxial constraint of the specimen. When the specimen size is smaller ($B$ = 20 mm > $B$ = 18 mm), the side groove with the same proportional thickness has more significant influence on the specimen constraint. Therefore, the differences of J-integral fitting values of SENB18 specimens are greater than that of SENB20 specimens (as shown in Figure 7).

### 3.5. Fracture Toughness of Specimens

The side grooves size of the fracture toughness test specimen should follow a certain standard [31], that is, the bottom radius is $R = 0.4 \pm 0.2$ mm and the side grooves depth is $B\text{-}B_N = 0.2B \pm 1\%$ mm, which is shown in Figure 14a. However, the deviations of the bottom radius and side grooves depth are generated during the actual testing operation, which affects the determination of fracture toughness. Therefore, taking the SENB 20 specimen with side grooves as an example, the influence of side grooves on fracture toughness test is studied by changing the depth and bottom radius of side grooves. The geometric models of side grooves with different depth and bottom radiuses of SENB 20 specimens side grooves models with different sizes are shown in Figure 14.

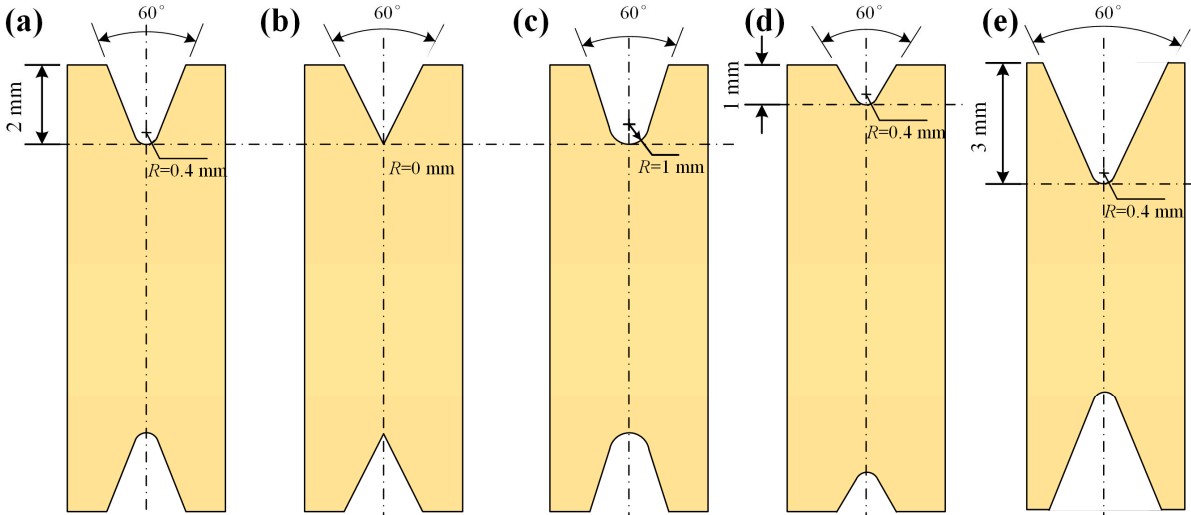

**Figure 14.** Geometric models of side grooves with different depth and bottom radiuses of SENB 20 specimens: (**a**) $R = 0.4$ mm, $B\text{-}B_N = 2$ mm; (**b**) $R = 0$ mm, $B\text{-}B_N = 2$ mm; (**c**) $R = 1$ mm, $B\text{-}B_N = 2$ mm; (**d**) $R = 0.4$ mm, $B\text{-}B_N = 1$ mm; (**e**) $R = 0.4$ mm, $B\text{-}B_N = 3$ mm.

The fracture toughness at crack initiation of five SENB 20 models with different depth and bottom radiuses are obtained by finite element simulation, which is shown in Figure 15. It can be seen that the fracture toughness calculated by different side grooves models are different. Compared with the standard side grooves specimen, if the side grooves depth remains unchanged, the fracture toughness decreases by 4.68% when the bottom radius decreases to 0 mm (Figure 14b), and it increases by 0.73% when the bottom radius increases to 1 mm (Figure 14c). While if the bottom radius remains unchanged, the fracture toughness increases by 3.33% when the side grooves depth decreases to 1 mm (Figure 14d), and it decreases by 3.85% when the side grooves depth increases to 3 mm (Figure 14e). It can be concluded that the deviation of the side grooves size can affect the determination of fracture toughness of the specimen.

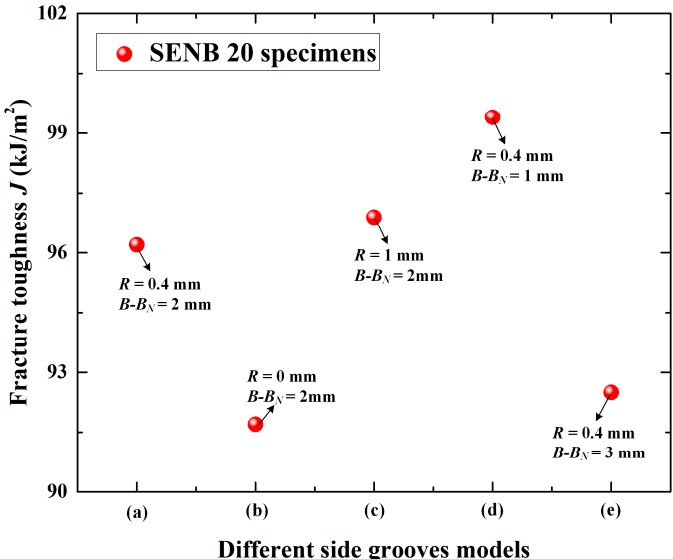

**Figure 15.** Fracture toughness at crack initiation of five SENB 20 models with different depth and bottom radiuses.

## 4. Conclusions

In order to investigate the stress triaxial constraint and fracture toughness properties of X90 pipeline steel, the experimental rules with different side grooves are proposed using the standards experiment and the corresponding numerical models are established in this paper. The resistance curves and fracture toughness values $J_{Q0.2BL}$ of each type of specimens are obtained and compared with the results of finite element analysis. Further, the stress distribution, J-integral distribution and stress triaxial constraint of the specimen are analyzed, as well as the influence of grooves size on the determination of fracture toughness is also discussed. The main conclusions are as follows:

(1) The resistance curves and fracture toughness are relatively different for the specimens without side grooves, while they are very close for the specimens with side grooves. Therefore, the side grooves specimen should be selected when measuring the resistance curve of X90 pipeline steel.

(2) The side grooves have an effect on the stress distribution, J-integral and stress triaxial constraint of the specimen. For the specimen without side grooves, the stress distribution, J-integral and stress triaxial constraint of the specimen reaches the maximum firstly in the thickness center and then begins to crack. However, for the specimen with side grooves, the stress distribution, J-integral and stress triaxial constraint of the specimen reaches the maximum firstly in the thickness edge and begins to crack, and the crack at the thickness center starts to crack later and propagates to both sides.

(3) The deviation of the side grooves size can affect the determination of fracture toughness of the specimen. When the side grooves depth remains unchanged, the fracture toughness decreases with the decreasing of root radius and increases with the increasing of root radius. When the root radius remains unchanged, the fracture toughness increases with the decreasing of side grooves depth and decreases with the increasing of side grooves depth.

**Author Contributions:** P.W., Methodology, Data curation, Validation, Writing-original draft; W.H., Investigation, Methodology, Validation, Writing—original draft; J.X., Formal analysis, Writing—review & editing; F.H., Conceptualization, Software, Validation, Writing—original draft; F.W., Methodology, Investigation, Validation, Writing—review & editing; C.H., Methodology, Formal analysis, Writing—review & editing. All authors have read and agreed to the published version of the manuscript.

**Funding:** This research was funded by National Natural Science Foundation of China, grant number 11572253 and 11972302.

**Institutional Review Board Statement:** Not applicable.

**Informed Consent Statement:** Not applicable.

**Data Availability Statement:** Not applicable.

**Acknowledgments:** All authors would like to gratefully acknowledge this support by National Natural Science Foundation of China.

**Conflicts of Interest:** The authors declare no conflict of interest.

## Nomenclature

| | |
|---|---|
| $W$ | height of specimen |
| $S$ | length of specimen |
| $B$ | thickness of specimen |
| $a_0$ | length of original fatigue crack |
| $B_N$ | effective thickness of the specimen with side grooves |
| $F_f^{1.3}$ | maximum fatigue crack prefabricated force when the crack growth is 1.3 mm |
| $F_f^{2.5\%W}$ | maximum fatigue crack prefabricated force when the crack growth is 2.5%$W$ |
| $F_f$ | Minimum of maximum fatigue crack prefabricated force |
| $g_1(a_0/W)$ | stress intensity factor coefficient |
| $R$ | loading stress ratio |
| $F$ | applied load |
| $V$ | notch opening displacement |
| $V_p$ | notch opening plastic displacement component |
| $A_p$ | area plastic component |
| $\eta_P$ | coefficient |
| $E$ | Young's modulus |
| $v$ | Poisson's ratio |
| $J_L$ | experimental equivalent to the J-integral values |
| $J_{ave}$ | J-integral average value |
| $J_{Q0.2BL}$ | fracture toughness value $J$ of the intersection of fitting curve and passivation line at 0.2 mm stable crack |
| $\alpha$ | fitting constant |
| $\beta$ | fitting constant |
| $R_m$ | tensile strength of material perpendicular to crack plane at test temperature |
| $\Delta a$ | stable crack growth including blunting |
| $\varepsilon_{nom}$ | nominal strain |
| $\sigma_{nom}$ | nominal stress |
| $\varepsilon$ | real strain |
| $\sigma$ | real stress |
| $T_z$ | stress triaxial constraints |
| $\sigma_{11}$ | normal stresses along x direction |
| $\sigma_{22}$ | normal stresses along y direction |
| $\sigma_{33}$ | normal stresses along z direction |

## Appendix A

The fatigue pre-crack of specimens should be prefabricated in the location of notch tip at room temperature [31]. The thickness $B$, height $W$ and effective thickness $B_N$ of specimen was measured with accuracy to ±0.02 mm or ±0.2%, whichever is larger. The thickness $B$ and effective thickness $B_N$ of specimen were measured before testing at a minimum of three equally-spaced positions along the intended crack extension path. The average of these measured results should be selected as the thickness $B$. For the specimen with side grooves, the effective thickness $B_N$ was measured before testing between the side grooves at a minimum of three equally-spaced positions along the intended crack extension path. The average of these measured results should be selected as the effective thickness $B_N$. The height $W$ should be measured at a minimum of three equally-spaced positions across the

specimen thickness on a line no further than 10% of the nominal height away from the crack plane. For all specimens with pre-crack, the ratio $a_0/W$ should be in the range of 0.45–0.70. The minimum fatigue crack extension should be the larger of 0.3 mm or 2.5%$W$. The bottom radius is $R = 0.4 \pm 0.2$ mm, angle is 60° and the side grooves depth is $B\text{-}B_N = 0.2B \pm 1\%$ mm. The corresponding fatigue crack prefabricated forces are given in Equations (1)–(3) [31]. Based on the maximum fatigue crack prefabricated force, the loading stress ratio was set to 0.1, the peak valley value was input and the frequency was set to 8 Hz to realize fatigue crack prefabrication. Then the length of pre-crack was measured to research the preset value (0.10$W$) in real time and the length of original fatigue crack is $a_0 = 0.45W + 0.10W$, as shown in Figure 4a. Three point bending fracture test was carried out after the prefabricated crack was completed, as shown in Figure 4b. The loading rate of the test was controlled to be 0.5 mm/min, the test data were collected every 0.1 s, and the software automatically recorded and saved the applied load and notch opening displacement. The displacement gauge should have an electrical output that represents notch opening displacement $V_p$ between precisely located gauge positions spanning the notch mouth. The verification of the displacement gauge should be performed at the temperature $\pm 5$ °C. The response of the displacement gauge should be accurate to $\pm 0.003$ mm and $\pm 1\%$ when displacement up to 0.3 mm.

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
