# Peer review of "Stress Triaxial Constraint and Fracture Toughness Properties of X90 Pipeline Steel"

_metals, doi:10.3390/met12010072_

Round 1
Reviewer 1 Report
Review
This paper presents the results of a study of the effect of side grooves on stress constrain and fracture toughness for X90 pipeline steel.
The results of experimental determination of JQ0.2BL for SENB specimens of thickness B = 20 and B =18 mm with and without side grooves are presented. The finite element method is used to obtain the values of local stress distribution for these types of specimens and the values of the J-integral. At the end of the article the results are presented containing the analysis of the influence of side grooves geometry (depth and radius at the grooves tip) on the magnitude of fracture toughness of the specimens.
According to the results of research, the authors:
- conclude that “the side grooves should be select when measuring the resistance curve of X90 pipeline steel”;
- indicate how exactly side grooves affect the stress distribution and the J-integral;
- comment data on the effect of depth and radius of side grooves on the magnitude of fracture toughness.
In general, the feasibility of using side grooves in determining Jс is substantiated in many works, and the method of their use is described, in particular, in the standard ASTME 1820-15a.
The author’s contribution to this problem may be considered as the analysis of the issue - how much effective is the use of side grooves for X90 pipeline steel. Results on the effect of depth and radius of side grooves on the value of fracture toughness of specimen are interesting; however, such combinations of grooves' depth and radius were selected that didn't enable to make a final conclusion as to which specific geometry of side grooves should be used in determining the fracture toughness of X90 steel specimens.
Remarks and comments:
- The Introduction section contains a lot of information that is not directly related to the purpose of the work. At the same time, there is no review of publications on the influence of side grooves on fracture toughness in general and for X90 steel, in particular. It is only said that «… there are few studies on the influence of side grooves on the…» No reference is made to the relevant standards.
- "True stress-true strain" curve is built to strain values not exceeding 0.06 (Fig. 5). In the local region ahead of the crack, the magnitude of local strains may reach much higher values. It is not clear from the text of the article how the values of equivalent stress were calculated for larger deformations.
- It is not clear from the text of the article how exactly the values of J0,2BLwere calculated, which are given in Table
- The paper should comment the results obtained on path-dependence of the J-integral for specimens with and without side grooves..
- There are no references in the text to the publications from which formulae (2), (9) and (4) are taken.
- It is advisable to add a Nomenclature to the article; the details of fatigue precracking should be included in Appendix and agreed with the standard ASTME 1820-15a. It should be explained why instead of root radius 0.5 ± 0.2 mm, the authors applied side grooves with a radius of 0.4 ± 0.2 mm.
- Table 2 twice shows the parameter "Density", which has different values of 692 g/cm3 і 0.88 g/cm 3. It is not clear what is "Parameters", which is equal to 13.
- Figure 5e and Table 5 duplicate each other (exhibit the same information).
The article may be accepted for publication accounting for the above remarks and comments.

Author Response
Dear Reviewer,
Thank you for your constructive comments concerning my manuscript entitled "Stress triaxial constraint and fracture toughness properties of X90 pipeline steel" (Manuscript ID: metals-1506590).
I have learned much from your comments, which are fair, encouraging and constructive. After carefully studying the comments and your letter, I have made corresponding major revisions. Revised portion are marked in red in the revised manuscript. Please see the attachment.
Thank you and best regards.
Yours sincerely,
Prof. Fenghui Wang

Reviewer 2 Report
Dear authors.
The work named Stress triaxial constraint and fracture toughness properties of X90 pipeline steel is a nice study about the material’s fracture behaviour under three bending test and for different boundary conditions (triaxial stress). In my opinion the work is well structured and well done. Congratulations to authors.
A great part of the work is related with Finite Element Method (FEM) results; such results seem to be well supported by previous experimental results, so here we have a guaranty very necessary in a study like this one (it should be mandatory, not all the published papers have such comparison, I don’t understand why, but this is another history!). The good concordance between experimental and numerical results is a good signal to believe that the numerical simulation by FEM is well done.
I have found a few minor remarks (non-scientific remarks).
- The table 2 must be revised.
- In line 384 I think that there is a mistake with the numbering of eq. 8 (maybe eq. 9?).
In my opinion, and it’s only my opinion, it would be a good idea to include real pictures of the fracture surfaces for the specimens of the Fig. 9, that is to say, to demonstrate that the stress distribution showed in Fig. 9 is in good concordance with the real fracture behaviour of the specimens due to the experimental text (the fracture begins here and propagates to…). I believe that to include this topic can contribute to enhance the quality of the work.
King regards.
Author Response

(The authors gave the same response as above.)

Round 2
Reviewer 1 Report
The revised article may be accepted for publication. It is only necessary to clarify the reply to the second remark. The authors misunderstood its essence, and in the revised version of the manuscript, instead of the "true stress vs. true strain" curve, they built the "engineering stress vs. strain" curve. The essence of the remark was for the authors to indicate what exactly dependence (formula) they used to extrapolate the "true stress vs. true strain" curve for strains greater than 6%, since strains at the crack tip can exceed this value.
